# Distribution and natural infection status of synantrophic triatomines (Hemiptera: Reduviidae), vectors of *Trypanosoma cruzi*, reveals new epidemiological scenarios for chagas disease in the Highlands of Colombia

Omar Cantillo-Barraza[1]*, Manuel Medina[2], Sara Zuluaga[1], María Isabel Blanco[3], Rodrigo Caro[2], Jeiczon Jaimes-Dueñez[4], Virgilio Beltrán[2], Samanta CC Xavier[5], Omar Triana-Chavez[1]

1 Grupo BCEI Universidad de Antioquia, Medellín, Antioquia, Colombia, 2 Programa de Control de Enfermedades Transmitidas por Vectores, Secretaría de Salud Departamental, Tunja, Boyacá, Colombia, 3 Laboratorio Departamental de Salud Pública de Boyacá, Tunja, Boyacá, Colombia, 4 Grupo de Investigación en Ciencias Animales (GRICA), Universidad Cooperativa de Colombia UCC, Bucaramanga, Colombia, 5 Laboratorio Biología de Tripanosomatídeos (Labtrip), Fundación Oswaldo Cruz, Rio de Janeiro, Brasil

* omarcantillo@gmail.com

## Abstract

### Introduction

Updating the distribution and natural infection status of triatomine bugs is critical for planning, prioritizing, and implementing strategies to control Chagas disease (CD), especially after vector reduction programs. After carrying out a control program, the Department of Boyaca contains the highest number of Colombian municipalities certified by PAHO to be free of intradomiciliary transmission of *Trypanosoma cruzi* by *Rhodnius prolixus*. The present work describes the spatial distribution, natural infection (NI), and molecular characterization of *T. cruzi* in synantrophic triatomines from the Department of Boyaca in 2017 and 2018.

### Materials and methods

An entomological survey was conducted in 52 municipalities in Boyaca known to have had previous infestations of triatomine bugs. Insects were collected through active searches carried out by technical personnel from the Secretary of Health and community members using Triatomine Collection Stations (PITs-acronym in Spanish). For evaluation of natural infection, triatomines were identified morphologically and grouped in pools of one to five individuals of the same species collected in the same household. DNA derived from the feces of each pool of insects was analyzed by PCR for the presence of *T. cruzi* using primers flanking the satellite DNA of the parasite. SL-IR primers were used to differentiate TCI from the other DTUs and to identify different genotypes. The distribution of the collected triatomines was analyzed to determine any vector hotspots using spatial recreation.

**Data Availability Statement:** All relevant data are within the manuscript and its Supporting Information files.

**Funding:** The authors(s) received no specific founding for this work.

**Competing interests:** The authors have declared that no competing interests exist.

## Results

A total of 670 triatomine bugs was collected, belonging to five species: *Triatoma dimidiata* (73.2%), *Triatoma venosa* (16.7%), *Panstrongylus geniculatus* (5.7%), *Rhodnius prolixus* (4.4%), and *Panstrongylus rufotuberculatus* (0.4%), from 29 of the 52 municipalities. In total, 71.6% of the bugs were collected within houses (intradomiciliary) and the rest around the houses (peridomiciliary). *Triatoma dimidiata* was the most widely distributed species and had the highest natural infection index (37.8%), followed by *T. venosa* and *P. geniculatus*. TcI was the only DTU found, with the TcI Dom genotype identified in 80% of positive samples and TcI sylvatic in the other insects. Spatial analysis showed clusters of *T. dimidiata* and *T. venosa* in the northeast and southwest regions of Boyaca.

## Conclusions

After some municipalities were certified free of natural transmission within houses (intradomiciliary transmission) of *T. cruzi* by *R. prolixus*, *T. dimidiata* has become the most prevalent vector present, and represents a significant risk of resurgent CD transmission. However, *T. venosa*, *P. geniculatus*, and *P. rufotuberculatus* also contribute to the increased risk of transmission. The presence of residual *R. prolixus* may undo the successes achieved through vector elimination programs. The molecular and spatial analysis used here allows us to identify areas with an ongoing threat of parasite transmission and improve entomological surveillance strategies.

### Author summary

Chagas disease is one of the most important tropical diseases in the Americas. Since 2010, Colombia has implemented programs to interrupt the intradomiciliary transmission of *T. cruzi* by *R. prolixus*, Colombia's primary vector. Boyaca, located in this country's central region, is one of the most endemic departments for Chagas disease. Control measures have been implemented, and the intradomiciliary transmission of *T. cruzi* by *R. prolixus* has been significantly reduced in 24 municipalities, according to PAHO certifications. Currently, the main risks in these certified municipalities are the presence of secondary vectors and certain ecological conditions favorable to triatomines. In the present study, we provide evidence that *T. dimidiata* and *T. venosa* have become the most common species. We observed a cluster of these infected species in the northeast and southwest regions of Boyaca. *P. geniculatus* and *P. rufotuberculatus* were present but less abundant, and residual *R. prolixus* populations remain in some certified-free municipalities. The overall natural infection index of secondary vectors was 31%, and spatial analysis identified priority areas for implementing surveillance and control actions. We identified *T. cruzi* in municipalities above 2,000 meters above sea level, which traditionally are considered non-endemic regions in Colombia. This suggested new epidemiological scenarios for Chagas disease transmission in Colombia's highlands and requires more studies to examine these scenarios.

## Introduction

Triatomines (Hemiptera: Reduviidae) are hematophagous insects that play an essential role as vectors of *Trypanosoma cruzi*, the causative agent of Chagas Disease (CD), which is considered the most important anthropozoonotic infection in Latin America [1]. The parasite *T. cruzi* presents tremendous genetic diversity and has been divided into six discrete typing units (DTUs, TcI to TcVI) [2], and other genotypes found in bats designed as TcBat [3]. Although the circulation of all DTUs has been described in Colombia, TcI is the most widely distributed DTU in this country [4], and which is then subdivided into two major genotypes associated with transmission cycles; domestic (TcI Dom) and sylvatic (TcI sylvatic) [5,6].

The subfamily Triatominae comprises five tribes, 18 genera, and 154 described species, of which the genera *Triatoma*, *Rhodnius*, and *Panstrongylus* contain the main vectors of *T. cruzi* to humans [7]. In Colombia, 26 triatomine species have been reported, and *Rhodnius prolixus* is considered the most important vector due to its infestation indices, high natural prevalence of infection and vectorial capacity [8,9], and its preference to live in human houses. This species has been the main target of vector control programs in Colombia, whose efforts have resulted in the interruption of within household (intradomiciliary) transmission of *T. cruzi* by *R. prolixus* in 63 municipalities in 6 departments [9–11]. However, other invasive species of the genus *Triatoma* and *Panstrongylus* have gained importance in recent years due to their presence in homes, natural infection rates with *T. cruzi*, and incrimination as vectors [12–14].

Colombia has a wide distribution of triatomine bugs reported from 465 municipalities, 95% of which are located below 2,000 meters above sea level (masl), and which are considered endemic for Chagas disease [15,16]. However, *T. cruzi*-infected secondary species, such as *Triatoma dimidiata* and *Panstrongylus geniculatus*, have been found in municipalities from not endemic areas in the departments of Boyaca, Santander, and Cundinamarca suggesting changes in Chagas disease epidemiology [9,17]. Nonetheless, the entomological indices related to the risk of transmission are still unknown for these highland zones. A similar situation has been described in other Andean countries, such as Ecuador, Peru, and Bolivia, where the secondary and sylvatic vectors, *Panstrongylus rufotuberculatus* and *P. geniculatus* that were initially restricted to the Amazon basin have been reported to colonize zones located between 2,000 to 3,300 m.a.s.l. [18–23].

The department of Boyaca, located in the center of the country, on the eastern Andean region, has 24 municipalities certified free from intradomiciliary transmission of *T. cruzi* by *R. prolixus*, making it the department with the largest number of certified *T. cruzi*-free cities in Colombia [10,11]. Nine triatomine species have been described, in Boyaca, of which five (*T. dimidiata*, *T. venosa*, *P. geniculatus*, *P. rufotuberculatus*, and *Rhodnius pictipes*) have been found infected with *T. cruzi*. This represents a risk of transmission in the municipalities where they are found [8,17]. Recently, CD cases have been reported in areas of Boyaca that do not have *R. prolixus* and likely involves secondary species. These reports identify the need to update entomological surveillance to include these other vectors after the successes of interruption programs to reduce intradomiciliary transmission of CD [13]. The present study aimed to analyze the spatial distribution, natural infection (NI), and molecular characterization of *T. cruzi* in synanthropic triatomines from Boyaca between 2017 and 2018.

## Results

### Collection of triatominae

A total of 670 triatomines belonging to five species (*T. dimidiata*, *T. venosa*, *P. geniculatus*, *R. prolixus*, and *P. rufotuberculatus*) were collected in 29 of 52 municipalities studied.

**Table 1. Entomological indexes in municipalities from Boyaca, from March 2017 to November 2018.**

| Province | Municipality | Triatomine collection type | Altitude (m.a.s.l.) | Number households visited | Number of infested houses | Total number triatomines collected | Infestation index (%) | Colonization index (%) | Density index | Crowding index |
|---|---|---|---|---|---|---|---|---|---|---|
| Norte | Boavita* | AS | 2,192 | 1,031 | 17 | 19 | 1.6 | 25 | 0.02 | 1.1 |
| | Covarachia* | AS | 2,327 | 705 | 4 | 4 | 0.5 | 25 | 0.01 | 1.0 |
| | San Mateo* | AS | 2,233 | 401 | 22 | 34 | 5.4 | 36.4 | 0.08 | 1.5 |
| | Sativa Norte | PITS | 2,608 | - | - | 6 | - | - | - | - |
| | Soatá* | AS | 1,988 | 763 | 140 | 324 | 18.3 | 50 | 0.42 | 2.3 |
| | Susacon* | AS | 2,487 | 326 | 11 | 22 | 7.6 | 27.3 | 0.07 | 0.9 |
| | Tipacoque* | AS | 1,871 | 834 | 22 | 41 | 2.6 | 40.9 | 0.05 | 1.9 |
| | La Uvita* | AS | 2,367 | 35 | 0 | 0 | - | - | - | - |
| Gutierrez | Cubara | AS | 357 | 220 | 3 | 25 | 1.3 | 66.6 | 0.11 | 8.3 |
| | El Espino* | AS | 3,321 | 610 | 7 | 9 | 1.1 | 14 | 0.01 | 1.3 |
| | Guacamayas* | AS | 2,202 | 310 | 5 | 8 | 1.6 | 20 | 0.03 | 1.6 |
| | Panqueba* | AS | 2,245 | 125 | 3 | 4 | 4 | 33,3 | 0.03 | 0.8 |
| Lengupa | Zetaquira* | AS | 1,670 | 1,257 | 15 | 20 | 1.1 | 6.66 | 0.02 | 1.3 |
| | Miraflores* | AS | 1,513 | 144 | 21 | 21 | 14.5 | 16.6 | 0.15 | 1.0 |
| Neira | Garagoa* | AS | 1,659 | 1,123 | 10 | 12 | 0.8 | 0 | 0.01 | 1.2 |
| | Pachavita | PITS | 1,995 | - | - | 11 | - | - | - | - |
| | Chinavita* | AS | 1,809 | 520 | 2 | 16 | 0.4 | 0 | 0.03 | 8.0 |
| Oriente | Sutatenza* | AS | 1,943 | 1,289 | 4 | 5 | 0.3 | 0 | 0 | 1.3 |
| | Tenza | PITS | 1,562 | - | - | 2 | - | - | - | - |
| | Guateque | PITS | 1,804 | - | - | 22 | - | - | - | - |
| | La Capilla | PITS | 1,756 | - | - | 8 | - | - | - | - |
| Valderrama | Socotá* | AS | 2,383 | 548 | 13 | 25 | 2.4 | 100 | 0.05 | 1.9 |
| Occidente | Briceño | PITS | 1,355 | - | - | 1 | - | - | - | - |
| | La Victoria | PITS | 1,478 | - | - | 1 | - | - | - | - |
| | Maripí | AS | 1,272 | 1,634 | 7 | 11 | 0.4 | 10 | 0.01 | 1.6 |
| | S. Pablo Borbur | PITS | 677 | - | - | 1 | - | - | - | - |
| Libertad | Labranzagrande* | AS | 1,114 | 1,092 | 4 | 7 | 0.4 | 14.2 | 0.01 | 1.8 |
| | Paya* | AS | 982 | 490(468) | 9 | 5 | 1.8 | 100 | 0.01 | 0.6 |
| | Pajarito* | AS | 793 | 485 | 0 | 0 | - | - | - | - |
| Ricaurte | Chitaraque* | AS | 1,569 | 1,280 | 3 | 3 | 0.2 | 0 | 0 | 1.0 |
| | Santana* | AS | 1,591 | 1,140 | 0 | 0 | - | - | - | - |
| | San Jose de Pare* | AS | 1,519 | 1,345 | - | 3 | - | - | - | - |
| | Moniquirá* | AS | 1,669 | 2,073 | 0 | 0 | - | - | - | - |
| | Togui* | AS | 1655 | 749 | 0 | 0 | - | - | – | - |
| Total | | | | 20,529 | 322 | 670 | 1.6 | 41.25 | 0.04 | 1.8 |

* municipality certified free of intradomiciliary transmission of *T. cruzi* by *R. prolixus*. (AS) active search and PITs. NC Not Calculated

Triatomines were found in 21 of 26 municipalities using active searches, but were found in only 8 of 26 (30,7%) municipalities where PITS were installed (Table 1).

However, 19 of 24 municipalities that had been certified free of intradomiciliary transmission of *T. cruzi* by *R. prolixus* had triatomines (Table 1). The total distributions of these insects were: *T. dimidiata* (73.2%, 491/670), *T. venosa* (16.7%, 112/670), *P. geniculatus* (5.7%, 34/670), *R. prolixus* (4.4%, 30/670), and *P. rufotuberculatus* (0.4%, 3/670) (Table 1 and Fig 1). Of all the triatomines collected, 23.1% (155/670) were obtained in 2017 and 76.9% (515/670) in 2018. Of

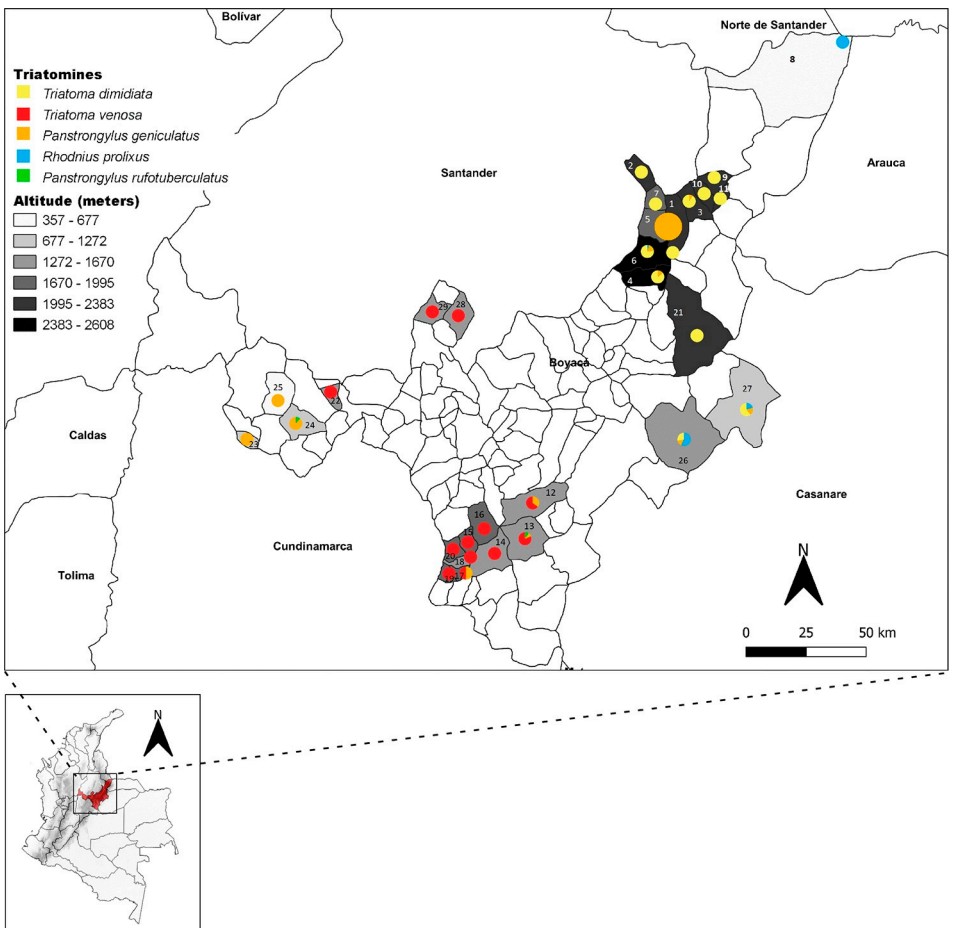

**Fig 1. Distributions of triatomines collected in the 29 infested municipalities in Boyaca Department (Colombia) between 2017 and 2018.** Grayscale box represent the altitudinal ranges (m.a.s.l.). Yellow circle - *T. dimidiata*, red circle - *T. venosa*, orange circle - *P. geniculatus*, blue circle - *R. prolixus* and green circle - *P. rufotuberculatus*. Altitude ranges are illustrated in grayscale. 1-Boavita, 2-Covarachia, 3-San Mateo, 4-Sativa Norte, 5-Soata, 6-Susacon, 7-Tipacoque, 8-Cubara, 9-El Espino, 10-Guacamayas, 11-Panqueba, 12-Zetaquira, 13-Miraflores, 14-Garagoa, 15-Pachavita, 16-Chinavita, 17-Sutatenza, 18-Tenza, 19-Guateque, 20- La Capilla, 21-Socota, 22-Briceño, 23-La Victoria, 24-Maripi, 25-San Pablo de Borbur, 26-Labranza Grande, 27-Paya, 28-Chirataque, 29-San Jose de Pare. The size of circles corresponds to number of triatomines recollected. Big circles more than 100 and small circles less than 100. The map was built using the free and open source QGIS software version 3.4 (https://www.qgis.org/en/site/forusers/download.html) and shapefiles were obtained from the free and open source DIVA-GIS site (https://www.diva-gis.org/gdata).

these, 65.2% (437/670) were adults, of which 71.4% (312/437) were found inside houses. Of the nymphs collected, 71.6% (167/233) were collected inside homes and the rest in peridomicile areas.

## Geographical distribution of triatomines

The most commonly collected species, *T. dimidiata* and *T. venosa*, were present in 44.8% (13/29) and 41.3% (12/29) respectively of the municipalities where triatomines were collected (Table 2). *Triatoma dimidiata* was found mainly in northwest provinces (Norte, Gutierrez, and Valderrama), in 50% (12/24) of the municipalities that had been certified to be free of intradomiciliary transmission of *T. cruzi* by *R. prolixus* (Table 2). Fifty-eight percent (285/491) of collected insects were adults, and the rest were nymphs. These were collected between 982

**Table 2. *T. cruzi* infection rates evaluated in triatomines collected in the 29 infested municipalities in Boyaca department (Colombia), from March 2017 to November 2018.**

| Province | Municipality | *T. dimidiata* | | *T. venosa* | | *P. geniculatus* | | *R. prolixus* | | *P. rufotuberculatus* | |
|---|---|---|---|---|---|---|---|---|---|---|---|
| | | Pools analyzed by PCR | Positive (%) | Pools analyzed by PCR | Positive (%) | Pools analyzed by PCR | Positive (%) | Pools analyzed by PCR | Positive (%) | Pools analyzed by PCR | Positive (%) |
| Norte | Boavita* | 13 | 4 (30.7) | - | - | - | - | - | - | - | - |
| | Covarachia* | 3 | 2 (66.6) | - | - | - | - | - | - | - | - |
| | San Mateo* | 22 | 11 (50) | - | - | 1 | 1 (100) | - | - | - | - |
| | Sativa Norte | 1 | 0 (0) | - | - | - | - | - | - | - | - |
| | Soata* | 185 | 66 (35.6) | - | - | 4 | 1 (25) | - | - | - | - |
| | Susacon* | 12 | 6 (50) | - | - | 2 | 1 (50) | - | - | - | - |
| | Tipacoque* | 30 | 13 (43.3) | - | - | - | - | - | - | - | - |
| Gutierrez | Cubara | - | - | - | - | - | - | 8 | 0 (0) | - | - |
| | El Espino* | 7 | 3 (42.8) | - | - | - | - | - | - | - | - |
| | Guacamayas* | 7 | 3 (42.8) | - | - | - | - | - | - | - | - |
| | Panqueba* | 2 | 1 (50) | - | - | - | - | - | - | - | - |
| Lengupa | Zetaquira* | - | - | 4 | 0 (0) | 7 | 3 (42.8) | - | - | - | - |
| | Miraflores* | - | - | 11 | 1 | 2 | 1 (50) | - | - | 2 | 1 (50) |
| Neira | Garagoa* | - | - | 11 | 0 (0) | - | - | - | - | - | - |
| | Pachavita | - | - | 5 | 0 (0) | - | - | - | - | - | - |
| | Chinavita* | - | - | 10 | 0 (0) | - | - | - | - | - | - |
| Oriente | Sutatenza* | - | - | 5 | 0 (0) | - | - | - | - | - | - |
| | Tenza | - | - | 2 | 0 (0) | - | - | - | - | - | - |
| | Guateque | - | - | 13 | 3 (23.1) | - | - | - | - | - | - |
| | La Capilla | - | - | 4 | 0 (0) | - | - | - | - | - | - |
| Valderrama | Socota* | 11 | 3 (27.2) | - | - | - | - | - | - | - | - |
| Occidente | Briceño | - | - | 1 | 0 (0) | - | - | - | - | - | - |
| | La Victoria | - | - | - | - | 1 | 0 (0) | - | - | - | - |
| | Maripi | - | - | - | - | 4 | 1 (25) | - | - | 1 | 0 (0) |
| | S. Pablo Borbur | - | - | - | - | 1 | 1 (100) | - | - | - | - |
| Libertad | Labranzagrande* | 1 | 0 (0) | - | - | 2 | 0 (0) | 1 | 0 (0) | - | - |
| | Paya* | 2 | 0 (0) | - | - | 1 | 0 (0) | 1 | 0 (0) | - | - |
| Ricaurte | Chitaraque* | - | - | 3 | 0 (0) | - | - | - | - | - | - |
| | San Jose de Pare | - | - | 3 | 0 (0) | - | - | - | - | - | - |
| | | 296 | 112 (37.8) | 72 | 4 (5.5) | 25 | 9 (36.0) | 10 | 0 (0) | 3 | 1 (33.3) |

and 2,608 m.a.s.l. (Tables 1 and 2). *Triatoma venosa* was found mainly in southwest provinces (Lengupa, Neira, and Oriente), and present in 29.1% (7/24) of the municipalities certified free of *R. prolixus* (Table 1), with 83% (93/112) of individuals collected being adults and the rest were nymphs. *T. venosa* was found at altitudes 1,355 to 1,995 m.a.s.l. (Tables 1 and 2 and Fig 1).

*Panstrongylus geniculatus*, *R. prolixus*, and *P. rufotuberculatus* were found in 34.4% (10/29), 10.3% (3/29), and 6.8% (2/29) of the infested municipalities, respectively, and at levels of 29.1% (7/24), 8.3% (2/24) and 4.1% (1/24) in municipalities certified to be free of intradomiciliary transmission by *R. prolixus*. The altitudinal range of these species was 677 to 2,487 m.a.s.l., 356 to 1,114 m.a.s.l., and 1,272 to 1,513 m.a.s.l., respectively (Tables 1 and 2 and Figs 1 and S1).

We found municipalities with concurrent infestations by two or three triatomine species. For instance, the presence of *T. dimidiata* and *P. geniculatus* (San Mateo, Sativa Norte, Soata and Susacon), *T. venosa* and *P. geniculatus* (Zetaquira), *P. geniculatus* and *P. rufotuberculatus* (Maripi), *T. dimidiata¸ R. prolixus* and *P. geniculatus* (Labranzagrande, Paya), *T. venosa*, *P. geniculatus* and *P. rufotuberculatus* (Miraflores) were observed (Table 2 and Fig 1).

## Entomological indices

The overall indices for all municipalities were an infestation index (1.6%), colonization index (41.25), density index (0.04), and crowding index (1.8) (Table 1 and Fig 2). The highest infestation indices were found in the municipality of Soata (18.3%), Miraflores (14.6%), Susacon (7.7%), San Mateo (5.5%), and Panqueba (4%) municipalities ($\chi^2$ = 943.6; $P$ < 0.05, df = 19), while the highest colonization index was found in Socota (100%), Paya (100%), Cubara (66.6%) and Soata (50%) ($\chi^2$ = 47.1; P < 0.05, df = 15), indicating that infestation rates and colonization indices throughout the municipalities are not homogeneous (Fig 2). The highest density indexes were found in Soata (0.42), Miraflores (0.15), and Cubara (0.11), while the highest crowding indexes were detected only in Cubara (8.3), Chinavita (8.0), and Soata (2.3) (Table 1 and Fig 2). In general, 72.6% (487/670) and 19.8% (133/670) of triatomine bugs were collected in the intradomicile and peridomicile areas, respectively, while the rest couldn't be assigned to a specific ecotope (Table 3). The species with the highest relative frequency inside houses were *T. dimidiata* (73.1%, 356/487), followed by *T. venosa* (16.4%, 80/487), *R. prolixus* (6.1%, 30/487), *P. geniculatus* (3.9%, 19/487), and *P. rufotuberculatus* (0.4%, 2/487). Meanwhile, *T. dimidiata* was the most commonly collected species in peridomicile areas (81.9%, 109/133), followed by *T. venosa* (9.0%, 12/133) and *R. prolixus* (9.0%, 12/133).

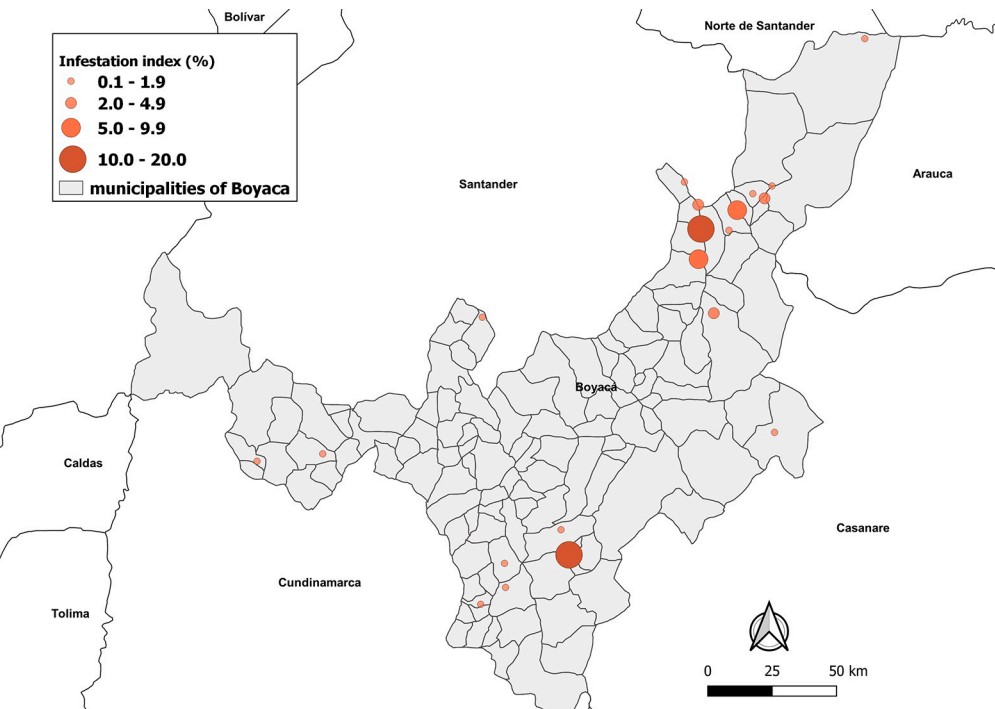

**Fig 2. Geographical distribution of Triatomine Infestation indices in Boyaca department of Colombia, from March 2017 to November 2018.** The map was built using the free and open source QGIS software version 3.4 (https://www.qgis.org/en/site/forusers/download.html) and shapefiles were obtained from the free and open source DIVA-GIS site (https://www.diva-gis.org/gdata).

**Table 3. *T. cruzi* infection rates evaluated in the triatomine species collected in intra- or peridomiciles in the 29 infested municipalities (Boyaca department, Colombia), from March 2017 to November 2018.** NT (Number of Triatomines)

| Species | Intradomicile | | | Peridomicile | | | Unknown origin | | | Total | | | $\chi^2$ | p-value |
|---|---|---|---|---|---|---|---|---|---|---|---|---|---|---|
| | NT | Pools analyzed by PCR | Positive (%) | NT | Pools analyzed by PCR | Positive (%) | NT | Pools analyzed by PCR | Positive (%) | NT | Pools analyzed by PCR | Positive (%) | | |
| *T. dimidiata* | 356 | 218 | 79 (36.2) | 109 | 57 | 22 (38.6) | 26 | 21 | 11 (52.4) | 491 | 296 | 112 (37.8) | 33.3 | <0.001 |
| *T. venosa* | 80 | 54 | 1 (1.8) | 12 | 10 | 0 (0) | 20 | 8 | 3 (37.5) | 112 | 72 | 4 (5.5) | | |
| *P. geniculatus* | 19 | 16 | 7 (43.7) | 12 | 6 | 2 (33.3) | 3 | 3 | 0 (0) | 34 | 25 | 9 (36) | | |
| *R. prolixus* | 30 | 10 | 0 (0) | - | - | - | - | - | - | 30 | 10 | 0 (0) | | |
| *P. rufotuberculatus* | 2 | 2 | 1 (50) | - | - | - | 1 | 1 | 0 (0) | 3 | 3 | 1 (33.3) | | |
| Total | 487 | 300 | 88 (29.4) | 133 | 73 | 24 (32.8) | 50 | 33 | 14 (42.4) | 670 | 406 | 126 (31) | | |

## Natural infection and ecotopes

A total of 406 triatomine pools were analyzed for *T. cruzi* infection, of which 73% (296/406), 17.7% (72/406), 6.1% (25/406), 2.4% (10/406), and 0.7% (3/406) correspond to *T. dimidiata*, *T. venosa*, *P. geniculatus*, *R. prolixus*, and *P. rufotuberculatus*, respectively. The overall natural infection index was 31% (126/406), with the highest values in peridomicile (32.8%; 24/73) compared to intradomicile (29.4%; 88/300) ($\chi^2 = 0.35$; $P > 0.05$, df = 1) (Table 3). At the species level, the highest infection index was found in *T. dimidiata* (37.8%; 112/296), followed by *P. geniculatus* (36%, 9/25), *P. rufotuberculatus* (33.3%, 1/3), and *T. venosa* (5.5%; 4/72), respectively ($\chi^2 = 33.3$; $P < 0.05$, df = 3). Interestingly, pools obtained from *R. prolixus* were not positive for *T. cruzi* infection. There were no significant differences in the infection indexes between each species' ecotopes (Table 3 and S1 Fig). The 188pb fragment was amplified in all PCR-negative samples, and inhibitors were not found.

## Genotyping of *T. cruzi*

Only TcI was found, and both genotypes (TcI Dom and TcI sylvatic) were detected (S2A and S2B Fig). 80% (49/61) of the analyzed sequences belonged to TcI Dom and 20% (12/61) to TcI sylvatic. 85% of the sample sequences from *T. dimidiata* corresponded to TcI Dom, while the rest was TcI sylvatic. Regarding *P. geniculatus*, an equal number of samples were TcIDom and TcI sylvatic (50% (3/6) each), and the sequences obtained from *P. rufotuberculatus* belong to the TcI sylvatic. We did not find a mixture of both genotypes. Geographically, TcI sylvatic was only located in Soata, El Espino, and Miraflores, while TcI Dom was present in eight municipalities (S2A and S2B Fig).

## Local indicators for spatial association

According to the global Moran´s Rate, there was a statistically significant spatial autocorrelation between triatomine distribution/infection indices (I = 0.467; $\leq 0.05$). The valvules obtained from the univariate Local Moran's index demonstrated a positive or direct spatial autocorrelation of *T. dimidiata* (I = 0.140; p-value $\leq 0.05$), *T. venosa* (I = 0.312; p-value $\leq$ 0.05) (Fig 3A and 3B), infection index (I = 0.135; p-value $\leq 0.05$) and infestation index (I = 0.118; p-value $\leq 0.05$) (Fig 3D and 3E). On the other hand, in relation to the genotypes, a spatial dependence was observed only in the distribution of TcI Dom (I = 0.240; p-value $\leq 0.05$) (Fig 3C). *P. geniculatus* (I = 0.04; p-value $\leq 0.05$), *R. prolixus* (I = -0.0039; p-value $\leq 0.05$) and *P. rufotuberculatus* (I = -0.017; p-value $\leq 0.05$) did not occur in clusters. Moreover, the

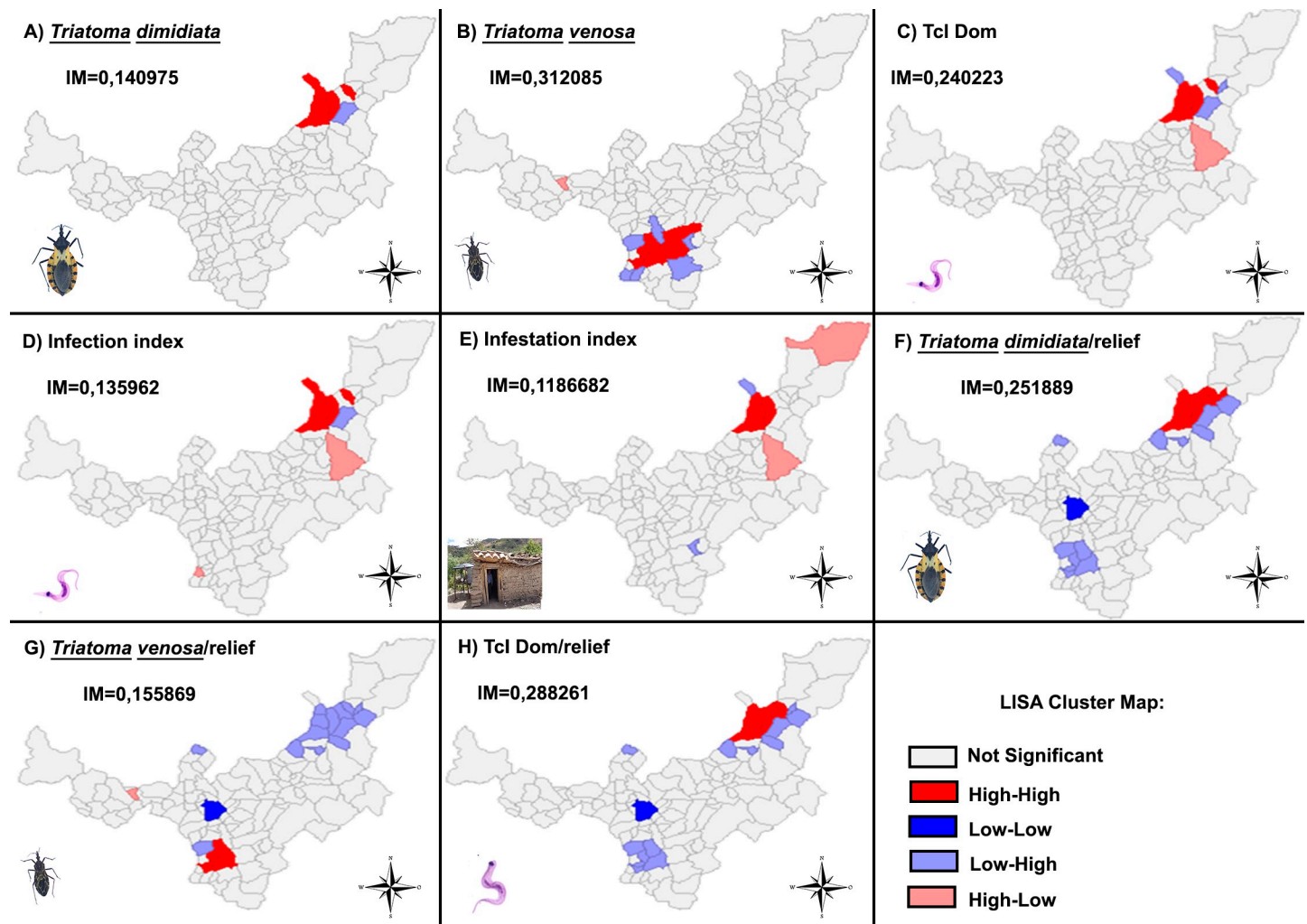

**Fig 3.** Maps of the index of correlation of univarite local Moran´s I in the distribution of: A. *T. dimidiata*, B. *T. venosa*, C. TcI Dom, D, Infection index, and E. Infestation index. Bivariate local Moran´s I between: F. *T. dimidiata*/relief, G. *T. venosa*/ relief and H TcIDom/ relief. Boyaca department between 2017 and 2018. The map was built using the free and open source QGIS software version 3.4 (https://www.qgis.org/en/site/forusers/download.html) and shapefiles were obtained from the free and open source DIVA-GIS site (https://www.diva-gis.org/gdata).

distribution of *T. dimidiata* (I = 0.251; p-value ≤ 0.05), *T. venosa* (I = 0.155; p-value ≤ 0.05) and TcI Dom (I = 0.288; p-value ≤ 0.05) (Fig 3F, 3G and 3H), were correlated with relief by bivariate Local Moran indices.

## Materials and methods

### Ethics statement

Ethical approval (Act No 113 of 2017) for analyzing animal species was obtained from the Antioquia University's animal ethics committee. All infested houses were sprayed with insecticide by SHB for the elimination of triatomine bugs.

### Study area

Boyaca is located in the center of Colombia, in the Andes: 04˚39'10" and 07˚03'17" North, 71˚ 57'49" and 74˚ 41'35" West. This department is made up of 123 municipalities, which are

divided into 13 provinces: Centro, Gutierrez, Lengupa, Libertad, Marquez, Neira, Norte, Occidente, Oriente, Ricaurte, Sugamuxi, Tundama, and Valderrama. The topography of this area, shaped by the eastern mountains, determines the presence of diverse landscapes and climatic conditions, from highlands (paramos) to hot climate areas (located less than 500 m.a.s.l.). This department has a mean annual temperature range of 22˚C to 9.7˚C and a mean annual rainfall of 645 mm. This study was carried out between March 2017 and November 2018 in 52 municipalities in nine provinces with a prior history of infestation with triatomines, according to the Boyaca Department Health Service (BDHS) (Table 1).

## Triatomine collection and processing

The 52 municipalities with historical reports of triatomine bugs in the department of Boyaca were divided into two groups: the first consisted of 24 municipalities certified to be free of intradomiciliary transmission of *T. cruzi* by *R. prolixus* in 2010 and 2017, and those that during the study period were developing activities to obtain the certification in 2019 [10,11]. Cubara and Maripi municipalities were included in this group even though they were not certified but had recent reports of *R. prolixus* (Table 1). For this group, one entomological sampling per year was carried out through active searches in the urban and rural areas of each municipality by trained agents from the Boyaca Department Health Service (BDHS) vector control program following the National Protocols of Entomological Surveillance [17]. In brief, synanthropic triatomines were sought in indoor and outdoor ecotopes for 30 min each; flashlights were used to look into cracks and crevices throughout the fabric of buildings walls, behind pictures of the walls, behind furniture, in closets, and especially, under bedding material. In addition, BDHS health workers left plastic pots in which householders were asked to collect any triatomines that they found. These insects were brought by householders to the Triatomine Recollection Stations (PITs-acronym in Spanish) established by BDHS to monitor the presence of triatomines. A total of 20,529 households were visited once every year during the study period (Table 1).

The second group consisted of the 26 remaining municipalities, where only the regular community surveillance was carried out (Table 1). Insects were collected in intradomicile and peridomicile environments by residents who sent them to the PITs of each municipality. All entomological samples sent to the PITS from all the municipalities were placed inside plastic containers labeled with the date, municipality, and site of collection (peridomicile or intradomicile), and then transported to the departmental public health laboratory where insects were identified using taxonomic keys [24].

## DNA extraction

Specimens were grouped into pools of one to five individuals based on their species and household of collection. For each pool, we dissected triatomines in a biological safety cabinet to avoid contamination. The feces were diluted in 300 μL of sterile PBS pH 7.2 and were used for DNA extractions. Genomic DNA was extracted from 200 μL of feces using the Genomic DNA purification kit (DNeasy Blood & Tissue kit Qiagen, Germantown, USA) following the manufacturer's instructions.

## Molecular detection of *T. cruzi* infection

All DNA preparations were screened for the presence of *T. cruzi* using a conventional PCR targeting satellite DNA [25,26]. The PCR was performed in a final volume of 25 μL containing 40–50 ng of genomic DNA, 1X of buffer, 0.04 mM of dNTP, 1.5 mM MgCl2, 0.4 μM of each primer (TCZ1 and TCZ2), and 0.05 U of Taq polymerase (Invitrogen, California, USA). The

thermal cycling conditions were as follows: pre-heating at 95˚C for 15 min, 40 cycles at 95˚C for 10 s, 55˚C for 15 s, and 72˚C for 10 s in a thermal cycler [25]. Amplification products were electrophoresed on a 1.5% agarose gel stained by ethidium bromide and visualized under UV light. Samples were considered positive for *T. cruzi* when a band of 188 bp was observed in the gel.

## Molecular characterization of *T. cruzi*

Positive *T. cruzi* samples were analyzed for molecular discrimination of TcI DTU of the other DTUs based on the amplification of the spliced leader intergenic region (SL-IR) gene using the primers TCC, TC1, and TC2, as previously reported [27,28]. The PCR was performed in a final volume of 25 µL containing 40–50 ng of genomic DNA, 1X of a buffer, 0.25 mM of dNTP, 2 mM MgCl2, 0.4 µM of each primer, and 0.05 U of Taq polymerase (Invitrogen, California, USA). The thermal cycling conditions were as follows: pre-heating at 94˚C for 5 min, 35 cycles at 94˚C for 30 s, 55˚C for 30 s, and 72˚C for 45 s in a thermal cycler, and a final extension at 72˚C for 5 min. We also used part (PCR directed to the SL-IR region only) of the algorithm implemented by Hernández et al. [29]. Amplification products were run on a 1.5% agarose gel, stained by ethidium bromide, and visualized under UV light. TcI was identified in the samples when a band of 350 bp was observed in the gel. For TcI positive samples, SL-IR PCR products were purified and sequenced at the Macrogen sequencing service, Seoul, South Korea. All nucleotide sequences were edited and aligned using CLUSTALW as implemented in BioEdit v.7.1.9 [30], and the microsatellite motif of the spliced leader gene (positions ranking between ~14 to ~40) was omitted as suggested [31]. The highest nucleotide identity value of the sequences based on optimal global pairwise alignments of each SL-RI sequence against reference strains reported for Colombia [5] was calculated in BioEdit v.7.1.9 [30].

## Detection of PCR inhibition

The presence of PCR inhibitors was examined in PCR-negative feces. In brief, 1 µL of DNA from negative feces were mixed with 10 ng of DNA from the Gal 61 *T. cruzi* strain in the same reaction tube to reference the amplification of satellite DNA of *T. cruzi*, as described above. The PCR was considered as inhibited when the 188 bp fragment was not detected in the gel.

## Data analysis

To estimate the risk of infection, the following entomological indices were determined in the municipalities of group one: the infestation index (number of houses infested by triatomines / number of houses examined x 100); the colonization index (houses with triatomine nymphs/ number of houses positive for triatomine × 100), the density index (number of triatomines captured / number of houses examined); the crowding index (number of triatomines captured / number of houses with triatomines) and natural infection index (number of infected pools/ total number of analyzed pools x 100) according to WHO (2010) [32]. Proportional comparisons were performed using the Chi-square test ($\chi^2$) with a $P < 0.05$ considered as significant. Data analyses were performed using SPSS v.18.0 statistical software.

## Geospatial analysis

For the map construction of the distribution of the triatomines, points of the samples localization were visualized in a Geographic Information System (GIS) in the Quantum GIS software version 3.4 Madeira (https://www.qgis.org/en/site/forusers/download.html), using the continental, national, and State boundaries (Shapefiles), extracted from the free and open source

DIVA-GIS site (https://www.diva-gis.org/gdata). Coordinates were recorded from the WGS 84 Datum (World Geodetic System 1984) geodetic coordinate system. Mapping of the triatomine distribution in the urban and rural area of the 29 municipalities was done from the study area. For this, the relative distribution of the collected and examined triatomine species, infection index, infestation index, and TcI Dom genotype and sylvatic distribution indicate the total number of the collected and examined specimens, whereas pies display the percentage of collected and examined species. The spatial data were analyzed in a Geographic Information System (GIS) platform using the open-source (Quantum GIS) QGIS (v.3.4 Madeira), a free and open-source geographic information system software.

## Local indicators for the spatial association: potential clusters of the distribution of Triatomine species, infestation and infection indices

The Global Moran's I index was calculated to verify spatial autocorrelations between the distribution of triatomine species, infection indices, and infestation indices to identify transmission hotspot areas. To identify localized clusters, the local Moran's I index was calculated. For the univariate Local Moran (five species of triatomine: *T. dimidiata*, *T. venosa*, *P. geniculatus*, *R. prolixus*, and *P. rufotuberculatus; T. cruzi* infection index and infestation index) bivariate (five species of triatomine, *T. cruzi* infection, genotypes, and infestation index x relief), defined the neighborhood matrix using the weights manager tool. The criterion adopted was the queen-type contiguity, with 1 value order of contiguity. Regions with common borders were considered neighboring. To index value significance demonstration, ie, the index value was not randomly obtained, the pseudo-significance test was performed with 999 permutations, and the 0.05 p-value was adopted. In this context, the Moran's I index tests the null hypothesis of spatial independence in the entire area (Moran's I = 0), generating a global I value potentially varying from −1 to +1: negative values indicate an inverse correlation (dispersion) while positive values suggest a direct correlation (clustering) [33,34]. Local Moran's I analysis was performed to determine local spatial association patterns. In addition to showing the dispersion diagram, as in IGM, the local index produces significance and cluster maps to determine the presence, or not, of spatial dependence. The Local and Global Moran's indices were calculated using GeoDa software (1.12) (GeoDa Center for Geospatial Analysis and Computation, Arizona State University, Tempe, AZ, USA).

## Discussion

After 24 municipalities in Boyaca were certified by PAHO to be free from intradomiciliary transmission of *T. cruzi* through programs aimed to eliminate *R. prolixus* [10,11,35], we carried out an entomological survey to characterize the spatial distribution of secondary vectors found inside household units (intra-and peridomiciliary environments) in this area. The potential risk of *T. cruzi* transmission by secondary vectors in municipalities located above 2,000 m.a.s.l. constitutes a new scenario that must be included in regional CD elimination programs. The entomological, molecular, and spatial analyses revealed that *T. dimidiata* and *T. venosa* are the most abundant vectors with clustered distributions. Residual populations of *R. prolixus* in these municipalities certified by the PAHO indicate that the vector elimination programs must be strengthened and continued. The distribution of species, their role as vectors of *T. cruzi*, and infestation and infection indexes will allow us to identify priority areas for subsequent surveillance and control measures.

*Triatoma dimidiata* has traditionally been considered a vector of secondary importance in Colombia [9,36]. However, recent studies in areas certified by PAHO to be free of intradomiciliary transmission of *T. cruzi* by *R. prolixus* that the epidemiological importance of this

species in parasite transmission may be increasing. *Triatoma dimidiata* was the most common (73.2%), most widely distributed (44.8%), and showed the highest levels of prevalence (37.8%) of triatomine species collected in Boyaca 2017 and 2018 (Fig 1 and Table 2). This species has moved into houses; 72.5% of specimens were collected inside homes, and it has high colonization and crowding indices in the northeast provinces (Norte, Gutierrez, and Valderrama). These data are supported by similar entomological indices reported in Capitanejo and Macaravita (Santander); and Soata and Tipacoque (Boyaca) between 2006 and 2008, using the same methodologies [37]. Finally, the spatial analysis showed that this species has a clustered distribution in the northeast (Norte province), and contributes to *T. cruzi* transmission (S1B Fig). *Triatoma dimidiata* is a species with major epidemiological importance in Mexico, Central America, Ecuador, and some departments of Colombia [38–42]. Moreover, an essential epidemiological role for this species has been reported in Guatemala and Nicaragua, where it has maintained *T. cruzi* transmission after *R. prolixus* eliminating [43].

*Triatoma dimidiata* is distributed throughout eleven countries and has been reported from a wide range of altitudes, ranging from 0 to 3,100 m.a.s.l. [23]. Most reports of *T. dimidiata* from Central America and Ecuador are from areas located below 1,800 m.a.s.l. [23,44]. Our survey and spatial analysis results suggest that a different situation seems to exist in Colombia because this species is of epidemiological importance in municipalities located above 2,000 m. a.s.l. Other authors have reported the infection of *T. dimidiata* with *Trypanosoma spp.* and *T. cruzi* in municipalities in Boyaca, Cundinamarca, and Santander located above this altitudinal range [9,17]. Our results indicate that *T. dimidiata* could become established as a major intradomiciliary vector of *T. cruzi*, but further studies must evaluate *T. cruzi* transmission to humans in these highland areas.

*Triatoma venosa* has been considered a secondary vector species in Colombia due to the frequency with which they are reported inside home dwellings and peridomicile areas [17]. This sylvatic vector is principally distributed in the Andean region at altitudes between 125 to 2,700m.a.s.l. [23]. It has been recorded in eight departments and 87 municipalities where it has been related to acute outbreaks of CD [9,17]. Here, we show that *T. venosa* is the second most widely distributed triatomine with natural infection of 5.5%, and with a cluster in their distribution in southwest provinces: Lengupa, Neira, and Oriente (Fig 3). In our study, 17% of *T. venosa* collected were nymphs, and 71.4% were collected in intradomiciliary areas. Considering that previously ecoepidemiological studies in Boavita (Boyaca) showed high mobility of species between peridomicile and intradomiciliary environments [45], the results in the current study could be a consequence of *T. venosa* mobility in the peridomicile area or a result of a peculiar domiciliation of this triatomine species in this department [9,17]. Therefore, we suggest that both scenarios should be explored.

*Panstrongylus geniculatus* has been considered recently as a relevant vector for *T. cruzi* due to its geographic distribution, its record of domiciliation, and association with outbreaks of oral *T. cruzi* infections in Colombia and Venezuela [20–22]. This species, reported in 18 countries, has a wide altitudinal range of 0 to 3,842 m.a.s.l. [23]. Here, we report its presence in ten municipalities in Boyaca, three of which are located above 2,000 m.a.s.l. and found this species infected with *T. cruzi*. However, we did not find a clustered distribution [9,17]. The overall prevalence of infection in this species was 36% (S1C Fig), which was lower than that detected in other endemic areas of Colombia (58.8%) [23]. Molecular genetics studies of the population of *P. geniculatus* showed that this species has a monophyletic origin and contains four clades with no eco-epidemiological differences between them [46]. In this sense, the population found in Boyaca could have the same epidemiological relevance as Venezuela or other Colombian regions where this species has been involved in oral transmission outbreaks [46].

TcI was the only DTU found in *P. geniculatus*. These results are contrary to reports in different Colombia regions, where *P. geniculatus* was infected with TcI, TcII, TcIII, TcIV, and TcV [14, 47,48]. These differences could be related to this species' blood-meal source in the other departments, where its capacity to adapt to different food sources has been reported [14]. Finally, the presence of both genotypes of TcI in this species suggests its role in transporting wild populations of the parasite into houses [13,49].

*Panstrongylus rufotuberculatus* is considered a sylvatic vector in Colombia, where it is present in eight departments and 26 municipalities [9,17]. This species is found in intradomiciliary and peridomiciliary areas in the lowland areas and above 3,600 m.a.s.l. in Bolivia [22]. *P. rufotuberculatus* has been incriminated as a CD vector in the Andean and coastal regions of Ecuador [50]. In contrast, in Colombia, it has been considered a major epidemiological risk factor in Amalfi (Antioquia), where it was reported as the second most common triatomine caught inside dwellings [51]. Here we report the presence of *P. rufotuberculatus* in two municipalities from the study area. However, this species has been reported in this department's other six cities (S1E Fig) [9,17]. These results, including its high infection rate and its exclusive detection inside homes, suggest a high potential transmission risk in municipalities where this vector is present. As far as we know, this is the first time that TcI genotypes of *T. cruzi* have been identified in *P. rufotuberculatus*, showing its participation in the sylvatic cycle.

*Rhodnius prolixus* has been considered for many years to be the most important vector of *T. cruzi* in Colombia. This species was introduced accidentally into Boyaca and has been reported in 26 municipalities of this department between 2002 to 2014 [17]. Here we show its presence in 10.3% of the cities (Cubara, Labranzagrande, and Paya), supporting the data on the success of control programs and the virtual elimination of this species. However, the vector's presence in two municipalities (Labranzagrande and Paya) that had been certified free of *R. prolixus*-transmitted *T. cruzi* endangers the overall success of CD control programs. Although *R. prolixus* in this study were not infected with *T. cruzi*, the high crowding index detected in municipalities where this was the only species found (Cubara) (Tables 1 and 2 and S1D Fig) demonstrates the remarkable capacity of this species to colonize and establish intradomiciliary populations in this region of Colombia [8]. Reinforcing insecticide spraying and vector surveillance in these municipalities are necessary to prevent vectorial transmission of *T. cruzi* in cities where residual populations of this vector were detected.

In Colombia, municipalities located at heights above 2,000 m.a.s.l. have traditionally been considered as non-endemic for CD [15,16]. The national protocol for entomological surveillance and vector control of CD suggests continued surveillance to identify triatomine vectors in areas below the 2,300 m.a.s.l. [52]. However, we found populations of *T. dimidiata* and *P. geniculatus* infected with *T. cruzi* in municipalities located higher than the cut-off altitude. The entomological indices and the presence of TcI Dom suggest a regular domestic cycle in these municipalities. We suggest expanding the altitudinal range to improve our understanding of the true entomological range of these vectors and future research to determine intradomiciliary transmission above 2,000 m.a.s.l. in Colombia. The presence of triatomines in highland municipalities may be related to climate change, as suggested for other triatomine species [53–55]. Further research about climate change effects should be studied for this species.

Several limitations of the present study should be acknowledged, including (i) we could not carry out an active search in all the municipalities with a history of triatomine infestation, thus preventing comparisons and homogeneous sampling (ii) the differences in triatomine collections between 2017 and 2018 were affected by administrative problems that did not allow the early incorporation of health workers of BDHS, affecting the number of operational personnel to collect insects and the activation of PRT. (iii) The blood sources of synanthropic triatomines have not yet been evaluated, and future trials should be carried out to understand the risk and

contact events between triatomines and residents. (iv) The present work was not accompanied by a seroprevalence study of the residents whose homes we collected *T. cruzi* positive insects. Even though BDHS has carried out screenings in school children under eight years of age at the municipality level, there are no data from residents outside the school ages. New serological studies targeted towards residents of infested homes should be carried out to have a broader picture of the secondary vectors' risk. Finally, further studies are needed in these municipalities to determine seasonal variations in the abundance of secondary vectors to prioritize the effectiveness of control programs.

The interruption of intradomiciliary *T. cruzi* transmission by *R. prolixus* in some areas of Colombia has been a major advance in CD control. However, this is an anthropozoonotic disease with transmission carried out by many vector species. The sustainability of successful management programs requires continuous evaluations of entomological data of all vectors that could exploit the available niche after the elimination of common triatomines such as *R. prolixus*. A sustained surveillance program is essential pre and post primary vector eradication efforts.

## Supporting information

**S1 Fig. Geographical distribution triatomine and infection by species Boyaca department of Colombia.** *T. dimidiata* (A), *T. venosa* (B), *P. geniculatus* (C), *P. rufotuberculatus* (D) and *R. prolixus* (E) from March 2017 to November 2018. The map was built using the free and open source QGIS software version 3.4 (https://www.qgis.org/en/site/forusers/download.html) and shapefiles were obtained from the free and open source DIVA-GIS site (https://www.diva-gis.org/gdata).
(TIF)

**S2 Fig. Geographical distribution of TcI DTU genotypes, in municipalities in Boyaca department (Colombia) between 2017 and 2018.** (A) TcIDom and TcI sylvatic. (B) TcIDom and TcI sylvatic by triatomines species. The map was built using the free and open source QGIS software version 3.4 (https://www.qgis.org/en/site/forusers/download.html) and shapefiles were obtained from the free and open source DIVA-GIS site (https://www.diva-gis.org/gdata).
(TIF)

## Acknowledgments

This study was carried out thanks to the agreement No. 001088 of 2016 and 352 of 2018 signed between the Health Secretary of the Department of Boyacá, and the University of Antioquia (Biology and Control of Infectious Diseases Group, BCEI). **To prof. Carl Lowenberger for the English edition.**

## Author Contributions

**Conceptualization:** Omar Cantillo-Barraza, Manuel Medina, Sara Zuluaga, Virgilio Beltrán, Samanta CC Xavier, Omar Triana-Chavez.

**Data curation:** Omar Cantillo-Barraza, María Isabel Blanco, Rodrigo Caro, Jeiczon Jaimes-Dueñez.

**Formal analysis:** Omar Cantillo-Barraza, Jeiczon Jaimes-Dueñez, Samanta CC Xavier.

**Funding acquisition:** Omar Cantillo-Barraza, Manuel Medina, Virgilio Beltrán, Omar Triana-Chavez.

**Investigation:** Omar Cantillo-Barraza.

**Methodology:** Omar Cantillo-Barraza, Manuel Medina, Sara Zuluaga, María Isabel Blanco, Rodrigo Caro, Samanta CC Xavier, Omar Triana-Chavez.

**Project administration:** Omar Cantillo-Barraza, Omar Triana-Chavez.

**Resources:** Omar Cantillo-Barraza.

**Software:** Jeiczon Jaimes-Dueñez, Samanta CC Xavier.

**Supervision:** Omar Cantillo-Barraza, Omar Triana-Chavez.

**Validation:** Omar Cantillo-Barraza, Omar Triana-Chavez.

**Visualization:** Omar Cantillo-Barraza, Samanta CC Xavier.

**Writing – original draft:** Omar Cantillo-Barraza.

**Writing – review & editing:** Omar Cantillo-Barraza, Jeiczon Jaimes-Dueñez, Omar Triana-Chavez.

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
