## [Decision Letter · Decision Letter 0]

16 Sep 2020

Dear Dr Cantillo,

Thank you very much for submitting your manuscript "Distribution and Natural infection of Synantrophic Triatomines (Hemiptera: Reduviidae) vectors of Trypanosoma cruzi reveals new epidemiological scenarios for Chagas disease in highlands of Colombia" for consideration at PLOS Neglected Tropical Diseases. As with all papers reviewed by the journal, your manuscript was reviewed by members of the editorial board and by several independent reviewers. In light of the reviews (below this email), we would like to invite the resubmission of a significantly-revised version that takes into account the reviewers' comments. 

We cannot make any decision about publication until we have seen the revised manuscript and your response to the reviewers' comments. Your revised manuscript is also likely to be sent to reviewers for further evaluation.

Sincerely,

Ricardo E. Gürtler

Associate Editor

Eric Dumonteil

Deputy Editor

Reviewer's Responses to Questions

**Key Review Criteria Required for Acceptance?**

**Methods**

-Are the objectives of the study clearly articulated with a clear testable hypothesis stated?

-Is the study design appropriate to address the stated objectives?

-Is the population clearly described and appropriate for the hypothesis being tested?

-Is the sample size sufficient to ensure adequate power to address the hypothesis being tested?

-Were correct statistical analysis used to support conclusions?

-Are there concerns about ethical or regulatory requirements being met?

Reviewer #1: The submitted article has a descriptive approach. The aim of the study is clear though the objectives could be refined for sake of clarity. Overall, the study design is appropriate to address the stated objectives. However, it is necessary to clarify the study population, give more details on the study design and some of the estimated indices. Additionally, I would suggest including a new PCR assay to differentiate all DTUs. Authors could better exploit the impressive amount of data collected with further spatial analysis. See my general comments.

Authors report having an approved protocol to analyze animal species.

Reviewer #2: The methods are in accordance with the objectives: the numbers of the samples are appropriate; there are no concerns about ehtical or regulatory requirements

Reviewer #3: see attached comments

**Results**

-Does the analysis presented match the analysis plan?

-Are the results clearly and completely presented?

-Are the figures (Tables, Images) of sufficient quality for clarity?

Reviewer #1: The results presented are in accordance with the planned analyzes. 

See my general comments.

Reviewer #2: The obtained results match with the analysis plan. Some reccomendations were made in the comments below.

Legends need review, also the quality of the figs need to be improved

Reviewer #3: see attached comments

**Conclusions**

-Are the conclusions supported by the data presented?

-Are the limitations of analysis clearly described?

-Do the authors discuss how these data can be helpful to advance our understanding of the topic under study?

-Is public health relevance addressed?

Reviewer #1: The discussion and conclusion could be improved by making a comparison of this problem with other countries, regions and vector species. The authors did not mention any limitations of the study. 

See my general comments.

Reviewer #2: Conclusions are supported by the data and a few corrections in the disscussion were recommended

Reviewer #3: see attached comments

**Editorial and Data Presentation Modifications?**

Reviewer #1: Please correct some typos, also please use italics for specific name and the correct word is "species" not "specie".

Please review the use of decimals throughout the work.

Please provide the necessary information in the figures to avoid having to go back to M&M.

Table 1. I think there is a mistake in the name of the column “Number of infected houses”. Please explain as footnotes what “-” means, the same to “Number household” column and “*” in municipality.

Reviewer #2: (No Response)

Reviewer #3: see attached comments

**Summary and General Comments**

Reviewer #1: The authors of the manuscript entitled "Distribution and Natural infection of Synantrophic Triatomines (Hemiptera: Reduviidae) vectors of Trypanosoma cruzi reveals new epidemiological scenarios for Chagas disease in highlands of Colombia" propose to describe the distribution of secondary vectors of Chagas disease after the quasi-elimination of the main vector in the Department of Boyaca, Colombia and thus assess the risk of vector transmission by them. The authors obtain infestation data and bugs from approximately 14,000 houses, using two different collection methods. In addition, authors evaluate the infection status, the frequency of T. cruzi genotypes, and map all this information at the municipality level. This topic was also described in seven other departments of Colombia by Hernandez et al. (2016) and I think it could be suggested as a widespread pattern in this country. I believe that the collected data and in part the proposed analysis have strong potential from the point of view of public health for the country and Chagas disease research in general. However, there are a number of issues that need to be addressed or clarified before publication. 

Introduction: 

The authors could present a general aspects of the topic including bibliography on other countries and vector species. Also, include reference of the altitudinal distribution of the studied species.

M&M: 

Additional information on study area should be included. How were municipalities selected? There is not information on infestation by these vector species for the area before this study. It is not clear how many municipalities were certified by PAHO and the reference provided it does not seem to match.

The study design should be clarified. The 29 municipalities were divided into two groups according to the vector capture method, but it is not clear which municipalities was included in which group. It is necessary to clarify the entomological evaluation carried out for the second group and if it was used to estimate the infestation index despite the fact that the capture method differs between groups.

The authors estimate an infection index through the examination of a pool of triatomines collected in the same house. However, it is not clear how it was calculated. Justify why the authors decided to use this infection index and not an insect prevalence of infection. Please revise if the informed estimation is an index or a real rate as it is called in the discussion. 

The authors mention that they use the TCC-TC1-TC2 primers as described in Burgos et al. 2007 to discriminate TcI from the rest of the UDTs instead of Souto et al. 1996. This proposed protocol does not allow to differentiate TcI from TcIII or TcIV, because no amplification product is expected for the latest UDTs. In order to differentiate all UDTs I suggest reviewing the publication by Marcet el al. 2006., or reviewing again Burgos et al. 2007.

In my opinion, the authors could give more support to the results including new statistical analyzes to evaluate the potential aggregation or association of the distribution of insects, as well as of the infestation and infection index.

Line 351-352 says “Mapping of the triatomine distribution in the urban area of the 29 municipalities from the study area”. Were only urban houses mapped?

Why was the blood meal source not considered in this study? 

Results:

Please consider including other types of graphs, like bar ones, they will probably improve the comparison among vector species. 

Why do the authors estimate the infestation index by grouping all species despite different vectors could imply different risk for transmission?

Discussion: 

The discussion could be improved by including a more general overview of the problem and including comparisons with other studies involving the same or other vector species, both within Colombia and at the regional / global level.

The finding of T. dimidiata and P. geniculatus in municipalities located above their altitudinal range of distribution is very interesting as the authors pointed out. However, I found that in Guhl et al. 2007 (Reference 9) both species were previously found in these municipalities and were also found infected with T. cruzi. Please clarify this point.

How do the authors interpret the risk of transmission by secondary vectors suggested with the almost nil seroprevalence in humans under 18 years old in the study area? Does the seroprevalence study match spatially with the area of greatest infestation detected in this work?

The authors should include some limitations that they consider for the study.

There is a problem with the references cited. In some cases, it does not seem to support the text (eg Line 241: Cura et al. for altitudinal distribution of vector species; Line 200, references number 28 and 29; Line 243 Reference 36), and it seems that there is an error in the references order. I suggest reviewing them exhaustively. 

I suggest avoiding the writing of textually similar phrases from recent works published by the main author. For example the final sentence of the discussion and the abstract.

Reviewer #2: Review: Plos Tropical Neglected Diseases

Manuscript 

General Comments: The manuscript entitled “Distribution and Natural infection of Synantrophic Triatomines (Hemiptera: Reduviidae) vectors of Trypanosoma cruzi reveals new epidemiological scenarios for Chagas disease in highlands of Colombia” presents relevant information on several species vectors of Trypanosoma cruzi in the Boyaca department of Colombia. The subject of the manuscript is in accordance and in the scope of PTND. However, several corrections and a new revised version must be considered before its acceptance for publication.

Specific comments: Editorial corrections must be made throughout the manuscript.

English review is deeply recommended

Scientific names must be written in full in the beginning of a sentence or paragraph. Please check the scientific names throughout the manuscript. For example, check the lines 198, 215

Results / Discussion: Why a huge difference in the numbers of triatomines collected: “ Finally, 23.1% (155/670) of triatomines were collected in 2017 and 76.9% (515/670) in 2018.” This issue must be clarified and properly discussed

Line 107 specie should be corrected: species; not just in this specific line but, throughout the whole manuscript

Line 111 was founding should be corrected

Discussion

It would be interesting giving a broader view on the diversity of the Colombian triatomine vectors, for instance, mentioning how many species are recorded in the country, and how many of them are sylvatic however, eventually invade domiciles

Line 154 needs a reference

Review the sentence 170-180

Line 196 replace ”theory“ by “possibility”

Materials and Methods

Line 284- Replace the “niche” by “ecotopes”

References: need extensive review

Figs: Written information in the figs must be edited for better quality of the image

Legends: Tables 1-3 Should include ‘Colombia” all legends should be auto explanatory

Figures caption: should be revised

Reviewer #3: see attached comments

PLOS authors have the option to publish the peer review history of their article (what does this mean?). If published, this will include your full peer review and any attached files.

Reviewer #1: No

Reviewer #2: No

Reviewer #3: No
---

## [Decision Letter · Decision Letter 1]

14 Apr 2021

Dear Dr Cantillo,

Thank you very much for submitting your manuscript "Distribution and Natural Infection Status of Synantrophic Triatomines (Hemiptera: Reduviidae), Vectors of Trypanosoma cruzi, Reveals New Epidemiological Scenarios for Chagas Disease in the Highlands of Colombia" for consideration at PLOS Neglected Tropical Diseases. As with all papers reviewed by the journal, your manuscript was reviewed by members of the editorial board and by several independent reviewers. The reviewers appreciated the attention to an important topic. Based on the reviews, we are likely to accept this manuscript for publication, providing that you modify the manuscript according to the review recommendations. 

Sincerely,

Eric Dumonteil, Ph.D.

Deputy Editor

Eric Dumonteil

Deputy Editor

Reviewer's Responses to Questions

**Key Review Criteria Required for Acceptance?**

**Methods**

-Are the objectives of the study clearly articulated with a clear testable hypothesis stated?

-Is the study design appropriate to address the stated objectives?

-Is the population clearly described and appropriate for the hypothesis being tested?

-Is the sample size sufficient to ensure adequate power to address the hypothesis being tested?

-Were correct statistical analysis used to support conclusions?

-Are there concerns about ethical or regulatory requirements being met?

Reviewer #1: See Summary and General Comments

Reviewer #2: Objectives are clearly presented and M&M are pertinent and well described. The semple size is suficient. This second version is much better however, a few issues are still to be corrected.

**Results**

-Does the analysis presented match the analysis plan?

-Are the results clearly and completely presented?

-Are the figures (Tables, Images) of sufficient quality for clarity?

Reviewer #1: See Summary and General Comments

Reviewer #2: The analysis of the data presented are clear, figs and tabs are sufficient and in good quality

**Conclusions**

-Are the conclusions supported by the data presented?

-Are the limitations of analysis clearly described?

-Do the authors discuss how these data can be helpful to advance our understanding of the topic under study?

-Is public health relevance addressed?

Reviewer #1: See Summary and General Comments

Reviewer #2: Conclusions are in accordance to the obtained results and brings relevant recomendations.

**Editorial and Data Presentation Modifications?**

Reviewer #1: (No Response)

Reviewer #2: Please check general comments bellow.

**Summary and General Comments**

Reviewer #1: The authors made strides in the readability of the manuscript (ms) and overall ms has improved a lot. However, there are still several important corrections and some issues that need to be clarified before the ms can be accepted for publication.

Introduction

• I find somehow confusing how the certification of the municipalities from Boyacá department is named throughout the manuscript. Which is the correct epidemiological scenario: free of intra-domestic transmission; free of domestic R. prolixus; free of R. prolixus? Reference 24 refers to the certification of the interruption of the domiciliary transmission of Trypanosoma cruzi by Rhodnius prolixus. In the same way, please check if references 10 and 11 support the sentence on line 71-73. Is the statement “whose efforts have resulted in the certified elimination of domestic R. prolixus in 63 municipalities in 6 departments” correct?

Results

• The selection of municipalities remains a little confusing. In Mat&Met and at the beginning of the Results, the authors indicate that 52 municipalities were included in the study and were divided into two groups. However triatomines were found only in 29 municipalities. So, later in results the authors indicate that the municipalities studied are the 29 in which infestation was found. Is this correct? I would suggest including the municipalities from the first group in which no infestation was found for the calculations of the entomological indices.

• Line 106, please indicate what proportion of adult insects were found intradomicile.

• Line 146. I think the appropriate metric is "relative frequency" rather than "distribution", and it could be biased by the abundance of T. dimidiate. On the other hand, if the proportion of vectors of a given species found intradomicile was calculated, it would show that 100% of the R. prolixus were captured in this ecotope. Could this metric be more informative?

• Line 161, Table 3 and Fig SI The statistical test employed to evaluate significant differences is missing.

Discussion

• The ending of the first paragraph of the discussion should be improved. I do not understand how the diversity of triatomines will allow the identification of areas of action. Other named characteristics such as vector role in the transmission, species distribution, levels of infestation and infection could serve to stratify the areas according to risk of transmission.

• I suggest closing the manuscript with a paragraph of conclusions. For instance, the relevance of the herein presented results for the maintenance of transmission control in Colombia and other countries in similar situations could be included.

• Line 203, which are the previous levels of intra-domestic infestation by T. dimidiata in Boyaca or other places? Without this piece of information how can the authors support the hypothesize of “an advanced domiciliation process”? Could dwellers bias their catches towards this ecotope as has been recorded for T. infestans? 

• Line 220-222. The results of this work, although very interesting, suggest the participation of these vectors in domestic transmission. In my opinion new studies are needed to assess the risk of infection in humans.

• Line 265, the results presented "suggest" a high transmission risk. 

• Line 288-290, 

“We suggest expanding the altitudinal range to improve our understanding of the true entomological range of these vectors.”

It has already been reported that these species have a higher altitudinal distribution than 2000 masl (in Colombia and other countries). May be, what merits further research is to describe the domestic transmission above 2000 masl in Colombia.

• Line 299-303. What was the seroprevalence obtained by the 2019 BDHS study in children under 8 years of age?

M&M: 

• I think it could be useful to clarify the definition of "intra-domestic" and "domestic"

• There is something wrong in the expression of the inclusion of both municipalities (i.e. Maripi and Cubara). On the other hand, Table 1 does not show infestation data for R. prolixus

Minor comments

• Line 70 please replace with “natural infection rates”. Also, I think reference 9 is better for this sentence than reference 8.

• Line 77, I think the authors want to say “below 2,000 m.a.s.l.” 

• Line 101, “19 of these had been certified….(Table 1)” But in table 1 only 18 are marked with asterisk. Also, Maripi and Cubara are not marked with an asterisk. 

• Line 105, Table 2 does not indicate the number of collected triamones.

• Line 139, please eliminate the phrase “suggesting that infestation rates in the cities are not homogeneous”

• Line 140, the sentence “The highest colonization indexes were found in Socota (100), Paya (100), Cubara (66.6), and Soata (50.1)” repeats the information given in lines 136-137.

• Line 165, please replace for “Figure S2A and S2B”

• Line 166, please check the text font.

• Line 187, the word "transmission" is repeated.

• Line 189, consider using the term “distribution” rather than “concentration”

• Line 190, please include “The risk of T. cruzi by secondary vectors”

• Line 207, please rephrase as a conditional sentence.

• Line 354, please check the appropiateness of reference 20 in this sentence. Also for reference 28 (Ramirez 2010 must be included).

Reviewer #2: Plos NTD

Manuscript 

General Comments: 

The manuscript entitled: “ Distribution and Natural Infection Status of Synantrophic Triatomines (Hemiptera: Reduviidae), Vectors of Trypanosoma cruzi, Reveals New Epidemiological Scenarios for Chagas Disease in the Highlands of Colombia” has improved significantly. However, before its final acceptance a few corrections and changes must be carried out. I recommend a fine English and editorial review.

Introduction

Number of triatomine species is not updated. Please correct 150 + (References)

Line 70 – Correct “naturla” for “natural”

Results

Line 110 – Cross out “Triatomines”

Line 164 - Only TCI was found in positive samples. I would suggest cross out positive samples. 

Discussion

Line 186 - Data of the certification

Line 287 – Change “ suggest “ by “ recommend “

I am not sure that the authors carried the characterization of “spatial concentration” of secondary vectors since just sparce field captures were performed. I would say the authors recorded the presence of secondary vectors in 24 municipalities of Boyaca. Please recheck and use the most precise term for defining study carried out.

Line 202- Spaces are missing.

Lines 256, 257- “ This species has the ability to develop peridomicile and domestic colonies in both the lowland areas and above 3,600 m.a.s.l. in Bolivia [22]. This sentence need several corrections: I would suggest changing it for: This species can be found in domiciliary and peridomiciliary ecotopes in the lowland areas and also above 3,600 m.a.s.l. in Bolivia [22].

It seems to me that the term “ability” is very anthropomorphic. Also in the original sentence the correct would be domiciliary and peridomiciliary colonies. The term domestic is not the best term at all. Please check the meaning of domestic. The colony could be inside the house but the domestication process could not be necessarily involved. Use whenever possible the term “domiciliary” instead of “domestic”.

Lines 284-286 - Here, however, we found populations of T. dimidiata and P. geniculatus infected with T. cruzi in municipalities located higher than the recommended altitude. We would suggest using “previous recorded altitude.”

Caveats- Are well presented and discussed. I just would include a relevant topic related to the frequency of the field captures to check for the species populations oscillations along the year and seasons. 

M & M- I could not find an explicit information related to the criteria used for the insect identification. Please include a reference

PLOS authors have the option to publish the peer review history of their article (what does this mean?). If published, this will include your full peer review and any attached files.

Reviewer #1: No

Reviewer #2: No

Figure Files:

Data Requirements:

Reproducibility:

References

---

## [Editor Report · Decision Letter 2]

17 Jun 2021

Dear Dr Cantillo-Barraza,

We are pleased to inform you that your manuscript 'Distribution and Natural Infection Status of Synantrophic Triatomines (Hemiptera: Reduviidae), Vectors of Trypanosoma cruzi, Reveals New Epidemiological Scenarios for Chagas Disease in the Highlands of Colombia' has been provisionally accepted for publication in PLOS Neglected Tropical Diseases.

Best regards,

Ricardo E. Gürtler

Associate Editor

Eric Dumonteil

Deputy Editor

---

## [Editor Report · Acceptance letter]

13 Jul 2021

Dear Dr Cantillo-Barraza,

We are delighted to inform you that your manuscript, "Distribution and Natural Infection Status of Synantrophic Triatomines (Hemiptera: Reduviidae), Vectors of Trypanosoma cruzi, Reveals New Epidemiological Scenarios for Chagas Disease in the Highlands of Colombia," has been formally accepted for publication in PLOS Neglected Tropical Diseases.

Best regards,

Shaden Kamhawi

co-Editor-in-Chief

Paul Brindley

co-Editor-in-Chief
